# A Crisscrossing Competency Framework for Family–Preschool Partnerships: Perspectives from Chinese Kindergarten Teachers

**DOI:** 10.3390/bs15050694

**Published:** 2025-05-17

**Authors:** Pan Jiang, Xuhong Song, Qin Wang, Xiaomeng Wang, Fangbin Chen, Dongbo Tu

**Affiliations:** 1Jiangxi Normal University, Nanchang 330022, China; 202251200015@jxnu.edu.cn (P.J.); 005490@jxnu.edu.cn (Q.W.); 202451200013@jxnu.edu.cn (X.W.); chenfbin@jxnu.edu.cn (F.C.); 2School of Early Childhood Education, Shangrao Preschool Education College (Shangrao Campus), Shangrao 334099, China; 3Student Affairs Office, Jiangxi College of Foreign Studies (Yaohu Campus), Nanchang 330099, China; 202351200013@jxnu.edu.cn; 4Wenxin Academy, Henan Open University (Longzihu Campus), Zhengzhou 450046, China

**Keywords:** family–preschool partnerships (FPPs), crisscrossing competency framework, behavioral event interview, kindergarten teacher

## Abstract

The promotion of enhanced well-being among children and collaboration among families, schools, and communities is paramount and is a pressing concern in the global education sector. This necessitates that preschool teachers possess the necessary competencies for effective family-preschool partnerships (FPPs). This study explored the competencies necessary for Chinese kindergarten teachers to engage in FPP using behavioral event interviews with 30 participants. Thematic analysis identified key competency traits, and independent samples t-tests with Bonferroni correction compared collaboration competencies between outstanding and typical teachers, as well as across different career stages. Consequently, a comprehensive crisscrossing competency framework consisting of four quadrants was developed. This framework distinguishes between high-performance and general traits, as well as between stable and variable traits that may evolve across career stages. High-performance traits such as communication, expression, and relationship management should be prioritized in the training and recruitment of early childhood educators involved in FPP. In contrast, intrinsic qualities that foster successful FPP, such as child orientation, should be cultivated early and sustained throughout a teacher’s career. From a developmental perspective, this framework provides a crucial foundation for evaluating and training kindergarten teachers in the competencies essential for fostering effective FPP.

## 1. Introduction

While the involvement of families and communities in school education has often been overlooked or avoided in China, this issue is not unique to this context. Similar patterns have been observed in other countries, which can be attributed in part to the failure to recognize this topic as an integral component of every teacher’s professional responsibilities ([18]). Educational policy documents issued by the [40] ([40], [41]) highlight the importance of family–school–community cooperation, recognizing it as essential for fostering an environment conducive to children’s holistic development and well-being.

Collaboration among families, educational institutions, and society in the upbringing of children represents a pioneering advancement in educational ideology ([51]). [16]’s ([16]) “Overlapping Spheres of Influence” theory provides a framework that emphasizes the importance of trust and mutual commitment in fostering a successful family–school partnership. Such collaboration not only supports children’s academic achievements but also bolsters their overall well-being ([17]; [35]; [34]; [53]).

Nevertheless, the current practice of collaboration between teachers and parents in children’s education continues to face numerous challenges and obstacles. A significant factor contributing to this predicament is the deficiency in teachers’ “professionalism” and “competency” regarding family–school partnerships ([13]; [61]). In the actual implementation of collaborative efforts between teachers and parents, educators often lack sufficient competence to effectively guide and facilitate parental involvement in their children’s learning ([17]). Insufficient cognitive proficiency and a lack of relevant skills result in many teachers, especially those new to the profession, struggling to engage in fruitful communication and collaboration with their parents ([11]; [62]). Consequently, this deficiency in collaborative competence hampers teachers’ ability to effectively co-educate children in concert with their parents and the broader community ([55]). Hence, it is imperative that we direct our research focus to the enhancement of teachers’ competency for family–school partnerships. This study aims to address this need by examining the competencies required for preschool teachers to promote the development of a high-quality early childhood education workforce and contribute to children’s overall development.

## 2. Theoretical Background and Literature Review

### 2.1. Theoretical Background

#### 2.1.1. Competency Theory

Research on competency dates back to scientific management studies conducted by Taylor in the early 20th century, who is considered the father of management science. This exploration was initially termed the “Management Competency Movement” management competency movement ([47]). In 1973, McClelland published a paper titled “Testing for Competence Rather than for Intelligence” ([37]), where he argued for uncovering conditions and behavioral traits that genuinely impact an individual’s performance and explicitly introduced the concept of “competency.” Based on the perspectives derived from [39]’s ([39]) research, there are significant differences between high performers and average performers in terms of how they describe events and their approaches to dealing with them. The key characteristics that explain these differences were extracted. These characteristics are termed competencies because they are directly related to job performance and can distinguish high performers from average performers. Competency is defined as the amalgamation of individual characteristics required for effective performance across a spectrum of roles and work contexts, enabling successful completion of associated tasks ([7]).

#### 2.1.2. The Overlapping Spheres of Influence

[16]’s ([16]) conceptual framework of “overlapping domains” serves as the theoretical foundation for family–preschool partnerships. Drawing upon this theory, a growing body of literature emphasizes the pivotal role of fostering a collaborative partnership between parents and teachers, founded on trust and unwavering commitment, to facilitate optimal child development ([4]; [21]; [42]; [52]). This theory encompasses both internal and external structures (see Figure 1), with the area of overlap being influenced by both the teacher and parent. The overlap expands when parents and teachers share the same goals for the child and coordinate their activities effectively. Conversely, when contact is avoided or barriers are encountered, the two domains become more distinct and the overlap diminishes. Certainly, the interplay between the school and family fosters collaboration, as both entities bear the responsibility for the socialization of children. In recent years, the development of digital communication has further strengthened this collaboration. For example, digital platforms such as WeChat and Ding Talk have become valuable tools for enhancing communication between teachers and parents, enabling more timely, accessible, and continuous interaction.

### 2.2. Literature Review

In this study, the term “Family–School Partnerships”, which is widely recognized ([20]; [61]), has been adopted. Preschool Teachers’ Family–Preschool Partnerships Competency (FPPC) encompasses the essential knowledge, skills, beliefs, and attitudes that enable them to effectively leverage the potential of their joint efforts with parents. These competencies, referred to as FPPC, are crucial for establishing and nurturing a strong partnership between home and kindergarten settings, thereby fostering optimal child development.

Extant research predominantly emphasizes overarching teacher competencies ([36]; [66]), essential teaching-related competencies such as digital competencies ([27]), and a flipped learning approach ([3]). However, research on FPPC remains limited, with existing studies primarily exploring these competencies from the perspectives of specific stakeholder roles, as summarized in Table 1. This table synthesizes key FPPC frameworks, illustrating the varied conceptualizations based on different professional and contextual standpoints. For example, [60] ([60]) emphasizes relational, communicative, and contextual competencies within the parent–teacher interaction context in Norway. Similarly, [57] ([57]) highlight the importance of social and communication skills among preschool teachers in Sweden. [30] ([30]) explore active parental participation in Finland and Portugal, and [67] ([67]) and [70] ([70]) provide detailed frameworks in the Chinese context, emphasizing professional beliefs, ethics, and home–school collaboration. [5] ([5]) further expand on these themes by addressing the coordination of home education and continuous learning.

Despite these contributions, more exhaustive studies typically offer only a broad outline of the competence content. The FPPC frameworks that have been established tend to be universal and relatively static. Nevertheless, competency is a dynamic construct influenced by both personal characteristics and the external environment ([45]), and it can be both observed and measured, as well as developed ([50]). Furthermore, prior studies have incorporated factors from career development stages into competency frameworks to maintain adaptability and vitality in response to environmental changes ([44]). This approach ensures that competencies remain aligned with the evolving demands of the work environment. Existing research has not adequately focused on the discernibility and developmental aspects of the FPPC, leaving a critical gap in understanding how these competencies evolve and can be effectively nurtured.

This study aims to examine the structure and content of preschool teachers’ competencies in family–preschool partnerships (FPPCs) by identifying and analyzing key competency characteristics. The objectives are defined as follows:(1)Objective 1: To analyze the significant differences in competencies between outstanding and typical preschool teachers in family preschool partnership work.(2)Objective 2: To identify notable differences in competencies between early and late career stages in family preschool partnership work, exploring how these competencies evolve and develop with experience.(3)Objective 3: To develop a sustainable crisscrossing FPPC framework (a dual-axis design that categorizes competencies along two dimensions: the horizontal axis distinguishes between high-performance and general traits, while the vertical axis classifies these traits by their degree of variability). This framework aims to better support preschool teachers’ development and retention.

## 3. Methods and Materials

### 3.1. Methods

This study employed a mixed-methods approach that integrates both qualitative and quantitative research methods. Qualitative thematic analysis was applied to the data derived from behavioral event interviews, as outlined by [8] ([8]). Through an iterative process of inductive and deductive analysis, the necessary FPPC for kindergarten teachers was conceptualized. The qualitative analysis process encompassed the generation of an initial coding dictionary, search for themes, and scoring of FPPC based on different levels of proficiency. Quantitative analysis was conducted by performing independent sample t-tests on the mean FPPC scores of different groups of kindergarten teachers to identify significant differences in competencies between groups, with Bonferroni correction applied to control the family-wise error rate. The quantitative findings were further supplemented and validated by referencing verbatim responses from qualitative interviews. Figure 2 shows the research process diagram.

#### Behavioral Event Interview

[14] ([14]) contended that assessing participants’ abilities based solely on their articulated motivations and skills is unreliable; instead, they advocated for the utilization of individuals’ past actions and work experiences as a more accurate predictor of future job performance. Consequently, Shyr adapted the critical incident technique into behavioral event interviews, which have since emerged as a pivotal method for assessing competencies ([48]). This study utilized the Behavioral Event Interview methodology, culminating in the design of an interview framework and a foundational information form. The “Behavioral Event Interview Outline” encompasses not only the objectives and questions pertinent to the interview but also the procedural guidelines and critical considerations that should be observed throughout the interview process.

Interviewees were prompted to recall in detail 2–3 actual instances of family preschool partnerships they had recently encountered in kindergarten, encompassing both successful and unsuccessful cases. According to Interpersonal Theory ([28]), interviewees are required to provide succinct explanations when they perceive burdens and impediments to connections. The STAR (Situation/Task, Action, Result) method ([50]) was employed to assist the interviewer in conducting the dialogue. If kindergarten teachers cannot recall specific incidents related to perceived burdens and obstructed connections, hypothetical inquiries are posed concerning critical incidents following the same guidance as in the main discussion.

### 3.2. Samples

This study employed purposive sampling to select preschool teachers with relevant experience in FPP, followed by snowball sampling to expand the sample through participant referrals. A total of 30 preschool teachers from four provinces and municipalities in the east-central region of China were recruited for this study. The participants had a mean of ten years of experience in teaching kindergarten to children aged three to six years. Among these teachers, three worked in private (for-profit) kindergartens, whereas the remaining were employed in public (non-profit) kindergartens. Ten participants were taught in urban kindergartens, while the other twenty were taught in suburban areas. Participant information is presented in Table 2.

For Research Objective 1, this study employed the framework proposed by [7] ([7]). Participants were categorized based on their work performance into “outstanding” and “typical” groups. The classification of preschool teachers involved in FPP was determined through social evaluations (principals and colleagues) ([38]). Social evaluations were primarily conducted using a 5-point scale (a score of 1 represents the worst performance of the teacher’s family preschool partnership work, while a score of 5 represents the best performance of the job). Those with scores exceeding 3 points (i.e., 4 or 5 points) for the teacher’s family preschool partnerships efforts by one principal and two colleagues were deemed to be in the outstanding group. Ultimately, in accordance with recommendations from [50] ([50]), the study identified a total of 18 excellent teachers and 12 typical teachers, maintaining a ratio of 1.5:1.

For Research Objective 2, participants were grouped into early career stage and late career stage groups based on whether the kindergarten teachers had been engaged in family preschool partnerships for more than ten years. These findings indicate that, in the Chinese context, early childhood educators frequently use a decade of service as a critical threshold, beyond which significant differences in teachers’ psychological resilience and competencies are observed ([31]; [59]). Each group consisted of 15 participants, with a 1:1 ratio between the two groups.

For Research Objective 3 of developing a crisscrossing (two-dimensional) FPPC framework for preschool teachers, the differentiation between outstanding and typical traits identified in Objective 1 was utilized as the horizontal axis of the competency framework, whereas the traits that varied with career stage, identified in Objective 2, were utilized as the vertical axis of the competency framework.

### 3.3. Interview Procedure

Researchers use a voice recorder to capture the session, ensuring that they have obtained the interviewee’s consent. Simultaneously, they took notes using pen and paper to document key information throughout the interview process, which typically lasted approximately one hour. Finally, the recorded audio was transcribed into written text, and the transcriptions were reviewed by professionals in preschool education holding a master’s degree or higher to ensure accuracy and integrity.

### 3.4. Competency Coding

The research process adhered to the six phases of thematic analysis outlined by [8] ([8]): familiarizing oneself with the data, generating initial codes, searching for themes, reviewing codes, reviewing themes, and defining themes. Initially, a preliminary set of competencies was established based on an international literature review closely related to “Collaboration with parents” through the aggregation of similar or redundant items. Subsequently, two senior coders familiarized themselves with the interview data and generated an initial coding dictionary for kindergarten teachers’ competencies (see Appendix A Table A1 for reference). The two coders then searched for and confirmed the coding dictionary. After the sixth iteration, no new themes emerged, indicating that the study comprehensively covered the competencies for FPP. The finalized coding dictionary, detailed in Appendix A Table A2, comprises 20 competencies. These competencies were categorized into seven performance levels, ranging from 0 to 6, based on Bloom’s taxonomy of cognitive domains and competency classifications from prior studies ([71]). Each level was defined, including conceptual information related to each item and descriptive definitions corresponding to different performance levels. Finally, each competency level within the sample of 30 participants was scored by two independent coders on a scale from 0 to 6 based on both the frequency with which the competency appeared in the interviews ([6]) and the complexity of its demonstration ([50]). For example, teachers assess relationship management skills in the context of early childhood education when describing how they purposefully cultivate relationships with parents. Actions such as “initiating basic, individual interactions with parents” receive lower scores, while “implementing multiple (two or more) strategies to connect with parents” receive higher scores. Additionally, evaluation for “adopting complex and coherent approaches to establish trust with parents” receives the highest rating.

### 3.5. Data Analysis

Audio recordings from 30 formal interviews were transcribed into text, resulting in a total of 646,560 words. Utilizing NVivo14, two researchers with a master’s degree in education independently coded and rated the data. To ensure the integrity of the coding process, two text samples were randomly selected for coding without disclosing whether they belonged to the outstanding or typical group ([38]). Subsequently, a discussion of the coding standards was conducted to achieve consistency between the two coders. Preliminary analysis revealed that the competencies identified through both the average and the highest scores were fundamentally consistent. [71] ([71]) have mentioned that average scores demonstrated greater stability and differentiation; consequently, this study opted to utilize average rank scores for analysis. For instance, if the characteristic “proactiveness” was noted 3 times at score 4 and 5 times at score 3, then the total frequency for this characteristic would be 8, yielding an average score of $\frac{\left(3\times 4)+(5\times 3\right)}{3+5}$ = 3.85.

This study employed independent sample t-tests to examine differences in the average competency trait scores among the different groups of preschool teachers. Following the significant t-test results (*p* < 0.05), post hoc pairwise comparisons were conducted using the Bonferroni correction. To control for the family-wise error rate across the 20 hypothesis tests, the significance level was adjusted to α = 0.0025 (0.05/20) using the Bonferroni correction method. Effect sizes for pairwise comparisons were quantified using Cohen’s *d* ([22]), with absolute values interpreted as follows: 0.2 = small, 0.5 = medium and 0.8 = large ([9]). All statistical analyses were performed using SPSS version 26.0.

This study employed Category Agreement (CA) as a measure of coding reliability, which assesses consistency levels among multiple coders in their categorization of identical interview text content. The formula used for calculation is $CA=\frac{2S}{\left(T1+T2\right)}$, where *S* denotes the number of instances where the different coders categorized the content in the same manner, while *T*1 and *T*2 represent the total number of codes assigned by each coder, respectively. If a competency is found to differentiate between these two groups across samples of executives, it becomes part of a standardized dictionary of competencies ([38]).

## 4. Results

### 4.1. Coding Reliability

Using the above calculation method, the Category Agreement (CA) in this study was approximately 71%. Despite the limited sample size in this study, category validity approached the average level of judgmental agreement of 74% to 80% observed in competencies by proficient coders, as noted by [38] ([38]), which implies that the reliability of the coding procedure is deemed satisfactory.

### 4.2. Finding

The results pertaining to Objective 1 are listed in Table 3. Specifically, a significant difference was observed for “child orientation” (*t* = −4.444, *p* < 0.001, Cohen’s *d* = 1.656). This difference remained significant after the Bonferroni correction (Adj. *p* < 0.0025). The overall findings of Objective 1 are as follows: Kindergarten teachers in the outstanding group and typical group demonstrated significant differences across 10 items, including child orientation, recognition of importance, responsible follow-up, achievement motivation, emotional regulation, confidence, knowledge of family education, communication and expression, guidance and cooperation, and understanding and empathy, with all Adj. *p*s < 0.0025 and all Cohen’s *d*s > 0.5. This shows that these competency traits distinguish the outstanding group from the typical group. Conversely, no significant differences were observed for the following seven items: proactivity, differentiated teaching ability, knowledge of children’s physical and mental development, basic care and education knowledge, analysis and diagnosis, observation and understanding, planning and organization. In other words, there was no significant difference in these traits between the outstanding and typical groups. Notably, differences between groups for understanding and empathy, reflection and learning, and emotional regulation were not corrected by Bonferroni (Adj. *p* > 0.0025), but the raw significance level (*p* < 0.05) combined with a large effect size (all Cohen’s *d*s > 0.8) suggests potential differences. The current sample size may have limited the statistical power of the study. Therefore, qualitative research, such as teacher interviews, is recommended to complement the quantitative findings.

The results for Research Objective 2 are presented in Table 4. For instance, in terms of “relationship management”, a significant difference was observed (*t* = 4.293, *p* < 0.001, Cohen’s *d* = 1.595). This difference remained statistically significant after Bonferroni correction (Adj. *p* < 0.0025). The overall research findings of Research Objective 2 indicate that the early career stage group and the late career stage group exhibited significant differences in five traits: analysis and diagnosis, observation and understanding, planning and organization, individualized teaching ability, and relationship management (all Adj. *p* < 0.0025, and all Cohen’s *d*s > 0.8). This suggests that these characteristics are prone to change with variations in work experience. However, in terms of child orientation, confidence, knowledge of children’s physical and mental development, proactivity, reflective learning, positive attribution, achievement motivation, and emotional regulation, there were no significant differences between the two groups. Within the existing measurement framework and established career development pathways, the eight identified traits exhibited no statistically significant group differences, implying a potentially lower rate of change compared to the other dimensions. While recognition of importance (*p* = 0.008), responsible follow-up (*p* = 0.018), basic care and education knowledge (*p* = 0.021), knowledge of family education (*p* = 0.003), communication and expression (*p* = 0.008), guidance and cooperation (*p* = 0.047) and empathic perspective-taking skill (*p* = 0.014) did not survive Bonferroni correction (Adj. *p*s > 0.0025), the observed raw significance levels and large effect sizes (Cohen’s *d*s > 0.5) suggest potential practical significance. Given that the stringent Bonferroni correction for 20 tests may be overly conservative, qualitative research is recommended to complement these quantitative findings, and additional longitudinal studies are required to observe this stability.

Objective 3 is based on Objective 1 and Objective 2. Informed by [49]’s ([49]) multidimensional perspective of competencies, this study draws upon the results presented in Table 3 and Table 4, as well as the qualitative analysis of interview data. We integrated 20 competency traits into a newly developed competency matrix (see Figure 3), which contains two vertical and horizontal dimensions, forming a crisscrossing framework for FPPC.

The vertical dimension of the crisscrossing competency framework categorizes traits into high-performing and general traits, aligning with findings from previous competency studies ([23]; [50]). High-performing traits serve to distinguish the competency traits of high-performing teachers from those of average-performing teachers, exhibiting a degree of discernment and selection ability. In contrast, general traits encompass the foundational characteristics essential for kindergarten teachers engaged in FPP. The horizontal dimension of the intersecting competency framework reflects the changes in abilities across different career stages and is subdivided into stable traits and variable traits. Variable traits denote characteristics possessing significant potential for development as work experience increases, while stable traits refer to relatively unchanging characteristics intrinsically linked to the requisite skills for effective FPP. Variable traits are typically fostered through specialized curricula or ongoing professional development programs in kindergartens. Conversely, stable traits, often referred to as soft traits, encompass intrinsic drivers that influence behavior.

Quadrant I: High-Performance, Variable Traits

These are competencies that distinguish high-performing teachers and are likely to evolve with experience. They can be cultivated through targeted training and professional development. They are based on the results of the quantitative analyses of Objectives 1 and 2 and supplemented by further qualitative analyses of the interview data. Quadrant I includes guidance and collaboration, communication and expression, recognition of importance, responsibility follow-up, family education knowledge, empathic perspective-taking skill, and relationship management. These competencies empower teachers to establish and sustain strong relationships with parents, fostering a collaborative environment that supports student growth and well-being. Previous studies have underscored the significance of establishing and nurturing positive relationships with parents ([1]; [26]).

Communication and expression: The results revealed that communication and expression were identified as distinctive characteristics of competence in FPP, which characteristics varied depending on the stage of career performance development. Interviews with young kindergarten teachers revealed that while many educators are enthusiastic about discussing solutions to children’s issues, they often overlook the importance of effective communication and expression when engaging with parents. Frequent feedback focusing solely on problems may evoke feelings of annoyance and resistance from parents, fostering a negative and distrustful relationship. This dynamic can directly undermine the effectiveness of FPP.

Relationship management: This competency was classified within the first quadrant based on the results of quantitative analyses. Relationship management encompasses the cultivation and maintenance of relationships not only with parents but also among kindergarten teachers and their colleagues, particularly co-teachers. To gain parents’ trust and affirmation, kindergarten teachers must ensure that they uphold a consistent educational philosophy and attitude among themselves. This necessitates effective relationship management among collaborating teachers, as a unified approach is crucial for fostering parental confidence in the educational framework being implemented. Teacher Yu, a teacher with over 15 years of experience, shared the following in the interview:


*Through years of experience, I have come to understand the importance of effective relationship management. In our classroom, communication between the three teachers is vital. For instance, if an incident occurs, such as a child falling, my co-teacher and caregiver will immediately inform me. I always prioritize discussing such incidents with the child’s mother first. If I’m unaware of the situation when parents inquire, they may feel that the teachers are neglecting their child. That’s why communication and cooperation among teachers are essential. We also remind each other of various tasks to ensure nothing is forgotten. By maintaining a consistent approach when communicating with parents, we can take the initiative in these discussions.*
([68])

Recognition of the importance: This refers to a comprehensive and nuanced understanding of the significance of FPP in educational practice. Only by acknowledging the value of this partnership will teachers be motivated to take practical actions. Such collaborative endeavors profoundly influence the physical and emotional development of children. Consistent with previous research ([19]), educators generally hold positive attitudes toward the importance of parental involvement in early childhood education. This perception tends to be positively correlated with the duration of teachers’ professional experience. Interview data suggest that teachers increasingly recognize the significance of family collaboration as their years of experience and personal family life experiences grow. Later in her career, Teacher Guo emphasized that family–preschool partnerships should be viewed as a long-term, interactive process. This involves the continuous exchange of information, emotions, expectations, and ideas—an approach consistent with [2]’s ([2]) perspective. Teacher Guo explained the following during the interview:


*Currently, the communication of daily teaching tasks and the feedback provided in kindergartens account for only about one-fifth of the overall efforts dedicated to family–preschool partnerships, far less than half. In fact, the implementation of these partnerships spans the entire educational journey, beginning when a child enters kindergarten and continuing until they leave.*
([24])

Responsible follow-up: This competency involves the teacher’s ability to engage actively in FPP, guided by a strong sense of responsibility. This involves multiple, ongoing interactions with parents concerning children’s issues, rather than merely offering superficial feedback on basic educational challenges and task communication. Sustained interaction between teachers and parents has been identified as a critical component of collaborative relationships ([10]; [43]). Time and continuity are essential for building relationships that, in turn, facilitate communication ([56]). Qualitative analysis of interview data indicates that responsible follow-up, a skill developed through continuous professional experience, is a variable trait. Teacher Yu provided the following statement in our interview:


*Regardless of the form of home-school collaboration, it is a continuous process; it isn’t fragmented. It can extend from the lower grades all the way to the upper grades and even into elementary school. Communication with parents can persist throughout this journey. Therefore, at times, home-school collaboration requires patience. We need to allow ourselves time, give the children time, and also grant the parents time.*
([69])

Knowledge of family education: This competency emerges as a distinguishing trait essential for the effective implementation of FPP. Many teachers face challenges rooted in their own family backgrounds and limited professional experience. These difficulties are often compounded by the lack of systematic training in family education during their teacher preparation. As a result, they may have limited exposure to diverse family structures and parenting philosophies, which hinders their competence to engage effectively with families. When faced with unique family-related issues, this knowledge gap can leave them feeling powerless. Previous research ([30]) has indicated that parents perceive teachers as inadequately trained in family education, which significantly impairs their capacity to address challenges arising from varying family frameworks within the classroom. Parents express a desire for the education department to provide training programs that would equip teachers to better navigate these issues.

Quadrant II: High-Performance, Stable Traits

These stable competencies are intrinsic behavioral factors that remain relatively constant and are less influenced by career stages in FPP. These qualities subtly permeate the FPP process and can significantly impact parents’ trust and support for teachers through their behaviors. Included in this category are traits such as child-centeredness, emotional regulation, and self-confidence. Teacher Wu in our interview stated the following:


*Self-confidence and emotional stability have consistently characterized my approach to professional interactions. Even during initial communications with parents, I rarely experienced apprehension, consistently maintaining composure. Emotional stability is a critical attribute, particularly for educators; colleagues who are easily affected by parental interactions often exhibit broader emotional instability in their daily practice.*
([63])

Child-oriented approach: This competency emphasizes the importance of maintaining a child-centered approach throughout the home–school collaboration process, aiming to support and promote children’s healthy and joyful development. Previous studies indicate that many perspectives and requests from parents are centered on their individual children ([25]). But this trait, which necessitates value reconstruction rather than skills training, is relatively stable across career stages. This quality is also acknowledged as a critical competency for kindergarten teachers. Teacher Dong noted the following during the interview:


*When you genuinely care for a parent’s child, with true devotion and a commitment to guiding and accompanying the child at every stage in kindergarten, you foster a sense of love and respect that both the child and the parent can feel, leading to a positive perception of the teacher.*
([15])

Emotional regulation and self-confidence: These competencies refer to teachers’ abilities to effectively manage their emotions and maintain emotional stability. When addressing parents’ concerns, it is essential for teachers to convey confidence in their positive feedback and resist the urge to allow specific parental challenges to induce anxiety or trigger negative emotions. For instance, teacher Lv in our interview described the following:


*When dealing with parents, it’s essential for teachers to remain calm and communicate effectively; instead of reacting out of anger or frustration, I take the time to adjust my mindset and approach situations positively, even when I’m particularly upset.*
([33])

Quadrant III: General, Variable Traits

This quadrant captures stable, general traits that form the foundational elements of effective FPP and possess inherently stable characteristics. It includes knowledge of children’s physical and mental development, reflective learning, positive attribution, achievement motivation, and basic care and education knowledge. A notable point highlighted during interviews is that, due to encouragement and emphasis from Chinese educational departments, preschool teachers in China generally possess an understanding of and can actively engage in FPP. Consequently, the trait of proactivity does not exhibit significant differences between high-performing and general groups, which contrasts with previous domestic research ([67]) on preschool teachers’ FPP. Teacher Wang in our interview noted the following:


*Sometimes, for example, when issues arise in your classroom while you’re working, you should proactively reach out to parents to resolve these problems. This is a primary premise of our role as educators today.*
([58])

Achievement motivation, positive attribution, and reflective learning: These traits illustrate teachers’ commitment to effective home–school collaboration and their ongoing pursuit of successful outcomes. This intrinsic motivation inspires individuals to engage in actions aimed at specific goals, prompting them to reflect on and acquire the professional knowledge and skills necessary for successful collaboration. It encourages them to overcome growth bottlenecks, step outside their comfort zones, and achieve secondary growth. Combined with an in-depth analysis of the intrinsic impression factors on behavioral performance, such as the insufficient activation design of “achievement motivation” in the existing training system, this may constitute an obstacle, making it challenging to change these characteristics. Therefore, these intrinsic traits were classified as the general variable traits. This aligns with the findings of [50] ([50]), which highlighted that motivation can empower individuals with multiple goals to remain steadfast and persist in their efforts.

Quadrant IV: General, Stable Traits

This quadrant pertains to general traits that may vary, with analyses indicating significant differences based on years of experience in the role. This quadrant primarily encompasses observation and understanding, differentiated teaching ability, analytical and diagnostic skills and planning and organization. However, they are not definitive indicators of high performance but rather foundational traits that are amenable to change within preschool teachers’ FPP work.

## 5. Discussion

This study investigated the structural and developmental competencies of kindergarten teachers’ FPPC using a crisscrossing framework that integrated trait variability and performance relevance. The findings offer a more nuanced and developmental understanding of FPPC, transcending a static enumeration of competence and emphasizing the role and evolution of various traits in practice.

First, the study identified several high-performance traits, such as child orientation, emotional regulation, communication, and expression skills, directly influencing the effectiveness of family preschool partnerships. These results align with previous research indicating that relational and communication skills are essential for collaboration between teachers and parents ([60]). Given that these traits are cultivable, yet not universally present, schools should emphasize the precise identification of high-performance traits within their teacher management systems and deliver targeted training accordingly. Furthermore, the trainability of these traits underscores their potential for early identification and developmental interventions through structured assessments and interactive learning tools, including situational judgment tasks, mentoring, and video-based reflection ([46]; [12]).

Additionally, the analyses revealed that certain traits, such as reflective learning, positive attribution, and emotional stability, tend to remain relatively stable across different career stages. Since implicit trait policies are rooted in core beliefs and personality characteristics shaped by early socialization processes ([29]), their transformation often requires substantial time and effort. However, current teacher support and training systems often focus on superficial skill development, neglecting the deeper transformation of intrinsic traits ([64]). This may also reflect developmental stagnation in certain regions of China due to inadequate support for kindergarten teachers, rather than an inherent attribute of competence. Furthermore, foundational knowledge related to parenting and child development, both physical and psychological, was consistent between early- and late-career teachers. The findings contrast with [65]’s ([65]) observation that teachers’ competence in fostering FPP varies significantly across different developmental stages. A potential explanation lies in the enduring impact of pre-service theoretical training on Chinese kindergarten teachers coupled with insufficient post-service professional development to update their knowledge base ([64]).

To translate these findings into practice, our four-quadrant model can be applied to teacher training, evaluation, and professional development by differentiating competency traits based on performance orientation and variability. Quadrant I traits (high-performance and variable), such as empathy and communication skills, should be prioritized during recruitment and training through interactive skill instruction and feedback mechanisms. Quadrant II traits (high-performance, stable), such as emotional regulation and reflective thinking, are best identified through situational interviews during the teacher selection process ([32]) and require ongoing support to maintain stability. The third quadrant encompasses intrinsic tendencies, such as initiative and achievement motivation. Although these traits are not always overtly expressed in daily interactions, they can foster professional growth and resilience. Finally, Quadrant IV comprises foundational knowledge and basic skills; these traits (e.g., observation and understanding, differentiated teaching ability) tend to be developed through systematic training and practical work experience rather than relying solely on an individual’s predisposition or short-term experience. Therefore, it is important to pay attention to teachers’ performance in FPP activities in their daily work and to continuously monitor their growth and development in these areas.

## 6. Conclusions

Previous research has indicated that competencies serve as a vital link between educational preparation and job requirements ([54]). This study employs a mixed-methods approach, integrating qualitative and quantitative research techniques, to develop a comprehensive crisscrossing framework of FPPC. The framework not only distinguishes key high-performance traits from general traits pertinent to FPP but also conducts a comparative analysis of these traits based on the early career stage and late career stage, categorizing them as stable traits or variable traits. It is designed to aid in the training, development, and selection of teachers.

In this study, the competencies are organized into four quadrants, focusing on the high-performance traits found in the first and second quadrants. The first quadrant comprises high-performance and variable traits closely linked to knowledge and strategies essential for interpersonal communication, such as collaboration and communication skills. This indicates that teachers can continually refine these interpersonal strategies through personal experience and ongoing professional development, including systematic training courses or mentorship programs. The second quadrant consists of high-performance and stable traits, which are intrinsic qualities that facilitate successful FPP. These traits should be nurtured early in the pre-service teacher education phase. When selecting preschool teachers for roles in FPP after hiring, it is crucial to focus on candidates who already exhibit these high-performance traits.

## 7. Limitations and Future Research

Although this study offers a comprehensive understanding of kindergarten teachers’ FPPC using the trait-performance quadrant framework, several limitations warrant acknowledgment. The qualitative research design permits in-depth insights into the developmental logic and trait differences of the FPPC. However, the sample comprised only 30 kindergarten teachers from central China, representing a small and non-representative segment of early childhood educators nationwide. This limitation restricts the generalizability of the findings to other regions. Additionally, despite coder training and cross-verification, interpretation subjectivity may not have been completely eliminated. The cross-sectional nature of this study further constrains our ability to assess how FPPC traits evolve over time or in response to varying career stages and situational demands.

Future research should prioritize the use of large-scale, diverse samples to validate the proposed capability framework, encompassing teachers from more private and non-public institutions. Longitudinal designs would be particularly beneficial for tracking trait expression changes across different career stages. Furthermore, the development of standardized, context-sensitive assessment tools could facilitate the quantification of family and FPPC traits and inform recruitment and training programs. Finally, subsequent studies could investigate whether the identified capabilities can be transferred to other collaborative environments, such as school–community or school–healthcare partnerships, to further advance the field of early education.

## Figures and Tables

**Figure 1 behavsci-15-00694-f001:**
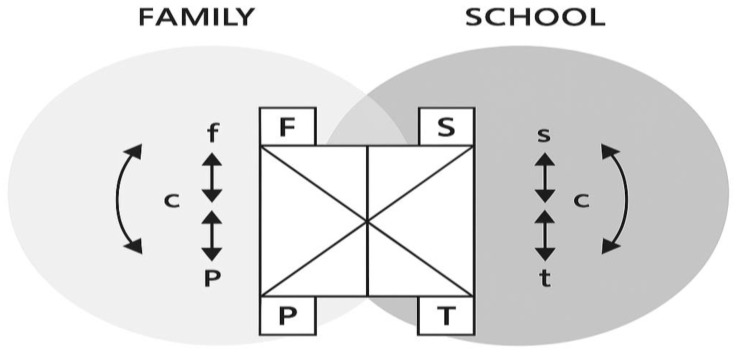
The overlapping spheres of influence of family, school, and community and their impact on children’s learning ([16]). Note: F/f = family, S/s = school, P/p = parent, T/t = teacher, c = community.

**Figure 2 behavsci-15-00694-f002:**
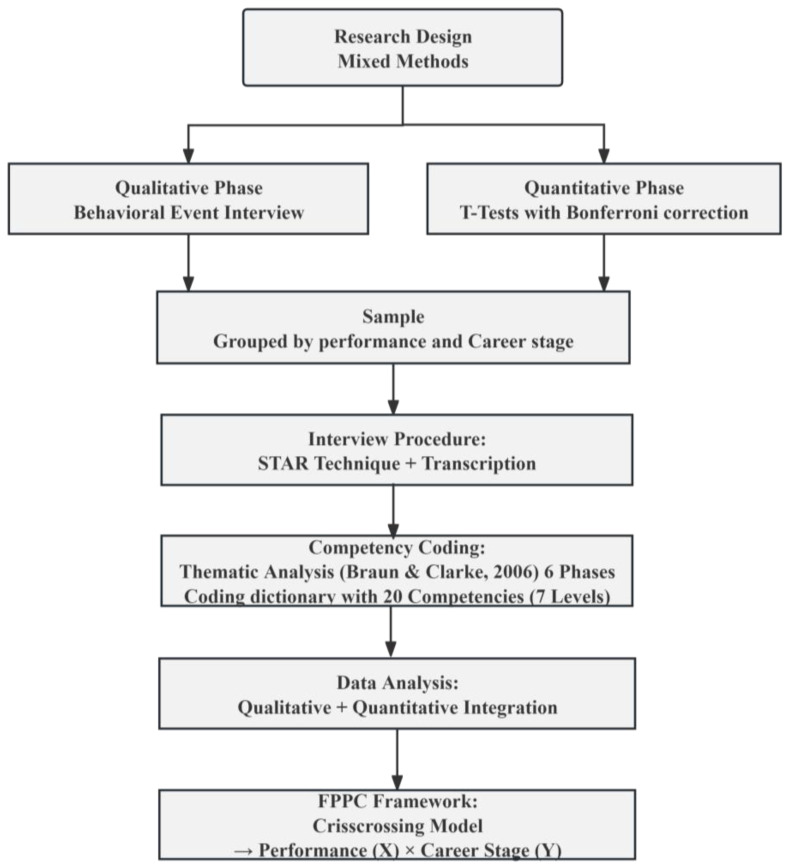
Research process diagram including thematic analysis ([8]).

**Figure 3 behavsci-15-00694-f003:**
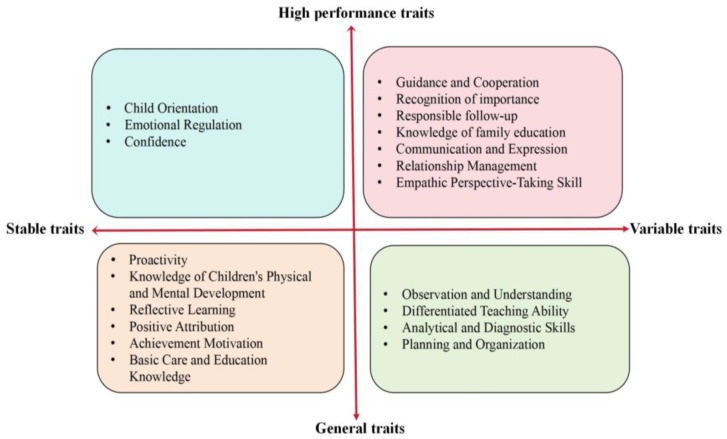
Crisscrossing model for FPPC.

**Table 1 behavsci-15-00694-t001:** Research on content of the FPPC framework.

Authors	Frameworks of FPPC	Perspectives of Various Roles
[60] ([60])	Relational competence, communication competence, contextual competence	Parents and teachers in Norway
[57] ([57])	Social and communication skills, the adept implementation of communicative teaching methodologies, capacity to effectively engage with parents	30 female preschool teachers in Sweden
[30] ([30])	Communication, professionalism, and invitations to active parental participation	Finnish (N = 10) and Portuguese (N = 9) parents
[67] ([67])	Professional beliefs and ethics, professional knowledge, and professional competence and divided into 10 secondary indicators	21 education experts in China
[70] ([70])	Homeschool partnership knowledge, homeschool partnership skills, attitudes and values, personality, and achievement motivation, which are organized into 21 indicators	Theoretical construction of the researcher
[5] ([5])	Ability to coordinate home education, ability to communicate with parents and teachers, ability to organize home education activities, ability to provide guidance on home education, ability to engage in continuous learning	Published literature in China

**Table 2 behavsci-15-00694-t002:** Participant information.

Sample	Gender	Experience Year	Kindergarten Type	Teaching Location
Teacher Zhu	Female	8	Public (non-profit)	Urban
Teacher Xu	Male	12	Public (non-profit)	Suburban
Teacher Dong	Female	20	Public (non-profit)	Suburban
Teacher Yu	Female	11	Public (non-profit)	Suburban
Teacher Yu	Female	14	Public (non-profit)	Suburban
Teacher Pan	Male	7	Public (non-profit)	Suburban
Teacher Li	Female	7	Public (non-profit)	Suburban
Teacher Yang	Female	20	Public (non-profit)	Suburban
Teacher Wang	Female	10	Public (non-profit)	Suburban
Teacher Guo	Female	7	Public (non-profit)	Urban
Teacher Zhang	Male	4	Public (non-profit)	Suburban
Teacher Mo	Female	15	Public (non-profit)	Urban
Teacher Bian	Male	14	Public (non-profit)	Urban
Teacher Sun	Female	8	Public (non-profit)	Suburban
Teacher Dong	Female	20	Public (non-profit)	Suburban
Teacher Shu	Female	6	Private (for-profit)	Urban
Teacher Lv	Female	11	Public (non-profit)	Suburban
Teacher Wu	Female	5	Private (for-profit)	Urban
Teacher Li	Female	6	Public (non-profit)	Suburban
Teacher Yao	Female	5	Public (non-profit)	Suburban
Teacher He	Male	5	Private (for-profit)	Urban
Teacher Wang	Male	8	Public (non-profit)	Suburban
Teacher Zhu	Female	5	Public (non-profit)	Suburban
Teacher Fu	Female	13	Public (non-profit)	Suburban
Teacher Chen	Female	14	Public (non-profit)	Suburban
Teacher You	Male	6	Public (non-profit)	Urban
Teacher Lou	Male	13	Public (non-profit)	Urban
Teacher Qiu	Female	5	Public (non-profit)	Urban
Teacher Ouyang	Female	11	Public (non-profit)	Suburban
Teacher Li	Female	11	Public (non-profit)	Suburban

**Table 3 behavsci-15-00694-t003:** Difference analysis between the standing and typical groups after Bonferroni correction.

Competency Items	Group (Mean ± Standard Deviation)		*t*	*p*	Adj. *p*	Cohen’s *d*
	Typical Group	Outstanding Group				
Child Oriented	3.00 ± 1.09	4.49 ± 0.75	−4.444	<0.000	**	1.656
Recognition of Importance	2.71 ± 0.96	4.56 ± 0.76	−5.887	<0.000	**	2.194
Responsible Follow-Up	3.27 ± 0.52	4.67 ± 0.77	−5.251	<0.000	**	2.032
Knowledge of Family Education	2.39 ± 0.96	4.03 ± 0.73	−5.314	<0.000	**	1.980
Confidence	2.05 ± 1.21	4.46 ± 0.95	−5.667	<0.000	**	2.285
Communication and Expression	2.75 ± 0.92	4.64 ± 0.58	−6.335	<0.000	**	2.587
Guidance and Cooperation	2.89 ± 0.80	4.44 ± 0.49	−5.996	<0.000	**	2.452
Relationship Management	2.57 ± 0.80	4.08 ± 0.90	−4.651	<0.000	**	1.754
Empathic Perspective-Taking Skill	2.60 ± 0.86	4.17 ± 0.72	−5.158	<0.000	**	2.034
Achievement Motivation	1.86 ± 0.81	3.94 ± 0.80	−6.942	<0.000	**	2.587
Reflective Learning	2.27 ± 0.96	3.66 ± 0.92	−3.818	0.001	0.020 #	1.477
Positive Attribution	2.65 ± 0.69	3.87 ± 1.14	−3.071	0.005	0.010 #	1.211
Emotional Regulation	3.03 ± 0.81	4.21 ± 1.05	−3.264	0.003	0.060 #	1.231
Proactivity	3.67 ± 0.89	4.34 ± 1.39	−1.474	0.152	—	0.549
Observation and Understanding	3.47 ± 1.11	3.63 ± 0.89	−0.41	0.685	—	0.157
Planning and Organization	3.17 ± 1.02	3.77 ± 1.33	−1.365	0.184	—	0.498
Differentiated Teaching Ability	3.82 ± 1.03	3.78 ± 1.02	0.091	0.928	—	0.034
Analytical and Diagnostic Skills	3.26 ± 1.23	3.56 ± 0.78	−0.812	0.424	—	0.302
Knowledge of Children’s Physical and Mental Development	3.68 ± 0.88	3.66 ± 1.31	0.043	0.966	—	0.016
Basic Care and Education Knowledge	3.50 ± 1.72	4.27 ± 0.95	−1.408	0.179	—	0.587

Note: ** Adj. *p* < 0.0025 (Bonferroni correction). # indicates significant before correction (*p* < 0.05) and Cohen’s *d* ≥ 0.8.

**Table 4 behavsci-15-00694-t004:** Difference analysis between the early career stage group and the late career stage group after Bonferroni correction.

Competency Items	Group (Mean ± Standard Deviation)		*t*	*p*	Adj. *p*	Cohen’s *d*
	Early Career Stage Group	Late Career Stage Group				
Relationship Management	4.01 ± 0.79	2.64 ± 0.92	4.293	<0.000	**	1.595
Observation and Understanding	4.31 ± 0.70	2.81 ± 0.51	6.471	<0.000	**	2.446
Planning and Organization	4.48 ± 0.82	2.60 ± 0.76	6.262	<0.000	**	2.367
Differentiated Teaching Ability	4.61 ± 0.51	2.93 ± 0.56	8.485	<0.000	**	3.153
Analytical and Diagnostic Skills	4.06 ± 0.72	2.83 ± 0.80	4.421	<0.000	**	1.614
Recognition of Importance	4.40 ± 1.15	3.24 ± 1.07	2.858	0.008	0.160 #	1.044
Responsible Follow-up	4.27 ± 1.00	3.41 ± 0.75	2.534	0.018	0.360 #	0.960
Basic Care and Education Knowledge	4.53 ± 0.53	3.39 ± 1.66	2.544	0.021	0.420 #	0.929
Knowledge of Family Education	3.98 ± 0.90	2.77 ± 1.08	3.311	0.003	0.060 #	1.209
Communication and Expression	4.31 ± 0.82	3.20 ± 1.25	2.874	0.008	0.160 #	1.049
Guidance and Cooperation	4.18 ± 0.84	3.46 ± 1.03	2.076	0.047	0.940 #	0.758
Empathic Perspective-Taking Skill	4.06 ± 0.88	3.09 ± 1.08	2.626	0.014	0.280 #	0.995
Reflective Learning	2.97 ± 0.77	3.26 ± 1.45	−0.675	0.507	—	0.255
Positive Attribution	3.40 ± 1.20	3.46 ± 1.15	−0.134	0.894	—	0.051
Proactivity	3.83 ± 1.30	4.31 ± 1.19	−1.068	0.294	—	0.390
Child Oriented	3.82 ± 1.30	3.97 ± 1.03	−0.355	0.725	—	0.130
Confidence	4.03 ± 1.13	2.86 ± 1.90	1.809	0.09	—	0.775
Knowledge of Children’s Physical and Mental Development	3.99 ± 1.02	3.37 ± 1.19	1.522	0.14	—	0.566
Achievement Motivation	2.96 ± 1.17	3.27 ± 1.45	−0.646	0.523	—	0.236
Emotional Regulation	3.75 ± 1.25	3.69 ± 1.00	0.146	0.885	—	0.054

Note: ** Adj. *p* < 0.0025 (Bonferroni correction). # indicates significant before correction (*p* < 0.05) and Cohen’s *d* ≥ 0.5.

## Data Availability

The original contributions presented in this study are included in the article. Further inquiries can be directed to the corresponding author.

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
