# Peer review of "A Crisscrossing Competency Framework for Family–Preschool Partnerships: Perspectives from Chinese Kindergarten Teachers"

_behavsci, 2025, doi:10.3390/bs15050694_

Round 1

Reviewer 1 Report

Comments and Suggestions for Authors

Overall, the contribution of this research to the journal is compelling, and the framework was developed with great detail. The structure, order, and coherence of the paper were good.

Introduction:

The article addresses the importance collaboration and partnership among schools, families, and communities for the overall quality of education for kindergarten students. The framework developed focuses on the competencies of teachers to determine effective characteristics. The introduction clearly described the problem and purpose statements for the study, presented well-defined theoretical foundations, and plainly identified the comparative groups of teachers. The three objectives were obviously stated.

Literature Review:

The reviewed literature contained multiple seminal pieces as well as a variety of relevant sources of application of the framework concepts. The literature referenced was supportive of the argument and framework justification.

Methods and Materials:

This section was very thorough. The discussion on the behavior event interview process was helpful in supporting the process of qualitative data collection and analysis.

Framework Development:

The framework was developed with a logical sequence and was well-supported by the literature and thoroughly discussed within the results and discussion sections.

Concerns:

The majority of concerns were in areas of grammar and formatting. The overall grammar was good. However, there were several subject/verb tense misalignments, a few possible missing words, and some basic capitalization errors. I am including a list below:

Line 27 – While needs to be capitalized.

Line 31 - In Chinese education system should be In the Chinese education system

Line 33 – Educational Official Documents  Should this be capitalized?

Line 51 - sufficient competence effectively  Should this be sufficient competence to effectively?

Line 110 – should be To analyze

Line 112 – should be To Identify

Line 206 – Use the word typical instead of normal for consistency.

Lines 295-305 – do not capitalize the item lists

Line 324 – do not capitalize child

Line 326 – tow should be two

Lines 329-334 – do not capitalize item lists

Line 403 – Recognition should be Recognizing

Line 621 – applicable should be application

Should these words really be bold?

Lines 69, 70, 77, 100 (top line of table and Table 1), 110, 112, 114, 121, 133, 183, 196, 208, 215, 235, 238, 295, 312 (top and bottom lines of table), 315, 339 (top line of table), 342, 365, 369, 447, 476, 503, 541, 559, 579 and 582

Formatting is off on quotes:

Lines 394-401, 416-418, 430-433, 453-456, 465-466, 473-474, and 487-488

Author Response

We would like to express our sincere gratitude for your thoughtful and constructive feedback. Thank you for your recognition of the structure, coherence, and theoretical contributions of our manuscript. Your positive comments gave us great encouragement and motivation to improve our work further. We are especially grateful for your detailed reading and careful attention to language and formatting. Your meticulous review, especially the attention to fine-grained linguistic and stylistic elements, reflects a high level of professional dedication and has been extremely helpful for refining our manuscript.

Please note that the page numbers referred to in this response letter correspond to the clean version of the revised manuscript.

Comments 1: “Line 27 – While needs to be capitalized.”

Response 1:Thank you for pointing this out. We have corrected the capitalization of "While" as suggested.

These changes can be found in Page 1, lines 30.

“While the involvement of families and communities”

Comments 2:" Line 31 - In Chinese education system should be In the Chinese education system."

Response 3: Yes, I agree that” In Chinese education system “should be” In the Chinese education system”. Thank you for catching this mistake. We have revised the sentence to improve grammatical correctness.

The modifications are located in the new sentence following the revised text on Page 1, lines 34-35.

Educational policy documents issued by the Chinese Ministry of Education (2023; 2024)”

Comments 3:" Line 33 – Educational Official Documents Should this be capitalized?

Response 3:Agrue, It is not a proper noun here and should not be capitalized. Thank you for your observation. We have corrected all item list formatting to remove improper capitalization.

The modifications are located in the new sentence following the revised text on Page 1, lines 34-35.

“Educational policy documents issued by the Chinese Ministry of Education (2023; 2024) highlight the importance…”

Comments 4:

“Line 51 - sufficient competence effectively  Should this be sufficient competence to effectively?”

Response 4:Thank you for catching this phrasing issue. We have revised the sentence to improve grammatical correctness.

Mention exactly where in the revised manuscript this change can be found Page 2, Line 50.

“sufficient competence to effectively…”

Comments 5:Line 110 – should be To analyze.”

Response 5: We appreciate your attention to detail. We have revised it to “To analyze”, and have also modified the subsequent sentence to “To identify, To develop” accordingly.

Mention exactly where in the revised manuscript this change can be found Page 4, Line 139,141,144.

Comments 6:Line 206 – Use the word typical instead of normal for consistency.”

Response 6: We appreciate the reviewer’s attention to terminological precision and consistency. The term “normal” has been revised to “typical” to more accurately convey the intended meaning.

Mention exactly where in the revised manuscript this change can be found Page 7, Line 207 .

Comments 7: “Lines 295-305 – do not capitalize the item lists”

“Line 324 – do not capitalize child”

“Lines 329-334 – do not capitalize item lists”

Response 7: We sincerely thank the reviewer for the detailed and careful feedback regarding the use of capitalization in item lists and terminology. Following the reviewer’s suggestions:

  • The item lists in lines 295–305 and lines 329–334 have been revised to lowercase.
  • The word “Child” has also been corrected to “child”.

These revisions can be found in the revised manuscript on Page9, lines 295–302 and Page 10, Line 314–330.

Comments 8: “Line 326 – tow should be two”

Response 8: Thank you for pointing out the typographical error in Line 326. We have corrected “tow” to “two” in the revised manuscript.

The correction can be found on Page10, Line 324 of the updated version.

“there were no significant differences between the two groups.”

Comments 9: “Line 403 – Recognition should be Recognizing”

Response 9: Thank you very much for your careful review and valuable suggestion. We appreciate your attention to the precise use of terminology and grammatical consistency. Indeed, “Recognizing” is typically used in sentence structures such as “Recognizing the importance of X is critical...”, where it serves as a gerund functioning as the subject of the sentence. Your point is linguistically accurate and well taken.

However, in this revised manuscript, we have intentionally used the noun form “Recognition” in this context because we are introducing and defining several core competency traits within our proposed framework. Specifically, we have fronted the key trait terms (e.g., Recognition of the importance) to make the structure more consistent and conceptually clear. This phrase is immediately followed by an explanatory sentence:

Recognition of the importance: This refers to a comprehensive and nuanced under-standing of the significance of FPP in educational practice.(Page12, Line401-402)

This formatting allows us to explicitly present and define each competence dimension as a distinct element of the framework. For this reason, we have retained “Recognition” rather than “Recognizing” to better reflect our intention and maintain consistency in presentation.

We sincerely appreciate your thoughtful feedback, which has helped us to further clarify our language choices.

Comments 10: “Should these words really be bold?

Lines 69, 70, 77, 100 (top line of table and Table 1), 110, 112, 114, 121, 133, 183, 196, 208, 215, 235, 238, 295, 312 (top and bottom lines of table), 315, 339 (top line of table), 342, 365, 369, 447, 476, 503, 541, 559, 579 and 582.”

Response 10: Thank you very much for this helpful reminder. Your comment has greatly enhanced my understanding of formatting conventions in SSCI journals. I now realize that bold formatting should only be used for structuring subheadings in the findings section and not for general emphasis. Based on your suggestion, I have carefully reviewed and revised all bolded expressions in the manuscript to ensure they are only used appropriately for section-level structuring. All necessary changes have been made accordingly.

These revisions can be found in the revised manuscript on lines 85, 97, 101-102, 112, 120,139, 141, 144, 162, 180, 194, 197, 208, 215, 235, 238, 290, 293, 313, 340, 343, 366.

In our current manuscript, both the table headers (e.g., Line 122, Table 1: ) and figure labels (e.g., Line 164, Figure 2: ) are presented in bold, and the top lines of tables are also bolded. This formatting was intentionally applied to help clearly distinguish the table headings from the data cells, thereby enhancing readability and visual clarity—an approach consistent with common practices in SSCI journal formatting.

Regarding the bolded phrases on lines 374, 383, 401, 419, and 434, our intention was to use them as subsection markers within the Findings section. Each bolded phrase represents a key competence dimension, followed by a colon and the corresponding explanation. This formatting helps structure the findings thematically and consistently. However, if the use of bold font in these cases is considered inappropriate, we are happy to revise them to regular or italic formatting as preferred.

We hope this explanation clarifies our decision, and we are grateful for your valuable input.

Comments 11:"Formatting is off on quotes: lines 394–401, 416–418, 430–433..."

Response 11: Thank you very much for your meticulous attention to the formatting of quotes. We appreciate your professional observation, which has prompted us to carefully review and revise all quoted interview content in accordance with APA 7 guidelines.

As you rightly pointed out, according to APA 7, direct quotations longer than 40 words should be formatted as block quotations, which requires:

  • Starting the quote on a new line
  • Omitting quotation marks
  • Placing any narrative lead-in (e.g., Teacher Guo notes:) outside the block
  • Citing the source at the end of the block or in the narrative

Accordingly, we have made the following changes:

Reformatted all quotes longer than 40 words (lines 394–401, 416–418, 430–433, 453–456, 465–466, 473–474, and 487–488) into APA-compliant block quotations.

Ensured each quotation is introduced clearly, with appropriate narrative framing preceding the block quote.

These revisions appear on lines 390–400, 413-419, 428-434, 452-457, 464-469, 474-478, 488-492 of the revised manuscript.

Example of revised formatting (Lines 413–419):

As teacher Guo explained during the interview:

Currently, the communication of daily teaching tasks and the feedback provided in kindergartens account for only about one-fifth of the overall efforts dedicated to family–preschool partnerships, far less than half. In fact, the implementation of these partnerships spans the entire educational journey, beginning when a child enters kindergarten and continuing until they leave (Teacher Guo, personal communication, February 2025).

We are grateful for your suggestion, which has significantly improved the formatting clarity and academic professionalism of our manuscript.

Final Note: Once again, we sincerely apologize for the language and formatting issues in the initial version. We have carefully considered your comments and made comprehensive revisions to improve grammatical accuracy and formatting consistency. All changes are clearly highlighted with yellow background shading in the revised manuscript.

 We have also carefully reviewed and revised the formatting of the entire manuscript and conducted a thorough language polishing. Detailed revisions can be found in the tracked changes version of the manuscript.

Thank you again for your insightful and detailed comments. We hope that the revised version of the manuscript meets your expectations and will be acceptable for publication.

Reviewer 2 Report

Comments and Suggestions for Authors

Dear author/s,

Thank you for giving me the opportunity to review the article.

A Crisscrossing Competency Framework for Family-Preschool

Partnerships: Perspectives from Chinese Kindergarten Teachers

The article does deal with an interesting, albeit not innovative, subject, but it is certainly important, and therefore consideration should be given to how to improve the article so that it can be submitted for re-judgment.

Abstract -  The abstract contains the main points of the article but does not account for the research process, which includes several participants, as well as a summary of the findings with recommendations from the research findings.

Keywords - It is advisable to use acronyms for phrases that will appear throughout the article, such as:  family-preschool partnerships=FPP

Introduction - According to the submitted article, there is no literature review, but on the other hand, the introduction to the work is very long. The introduction should be divided so that the article contains an introduction and a literature review.  Lines  26-153.

lines 75-80 : Correct wording

Figures 1 and 2 are not sharp and do not fit into the final version of the article.

Starting from line 117, there is an explanation of partnerships between the kindergarten and parents. There is no mention, for example, that nowadays communication with parents can be conducted via digital platforms such as WhatsApp for the benefit of the child's advancement and well-being.

Methods and Materials - A diagram describing the research process should be attached to this chapter, which will help the reader to concisely understand the research process in a consistent manner.

Results - is written in detail, but unfortunately due to technical problems it is very difficult to read. See for example lines 397-402, 412-414, 437-439, 462-464…….

This section should be re-edited so that it can be read clearly in its entirety.

Discussion - It is necessary to shorten and, above all, integrate and conduct a more extensive discussion between the review and the findings.

Conclusion - Well written and in a findings format.

Conclusions and limitations of the study should be separated into 2 separate subsections.

Limitations and Future Research - The text should be reduced by half, and the fact that this study included only 30 kindergarten teachers, which is a very small percentage of all kindergarten teachers in China and in relation to the size of the population, should be emphasized.

Also missing is a recommendation for further research on the subject, such as similar research with private kindergarten teachers and/or those not employed in public and/or government settings.

Line 790 unnecessary space delete

Lines 794, 797 & 812 right-aligned and reduced spacing.

I hope that the comments I have mentioned here will help the authors to improve their article.

With best wishes and success,

The Reviewer

Author Response

Thank you for your insightful and constructive comments. We greatly appreciate the time and effort you took to review our manuscript. Your feedback has been invaluable in improving the quality of this paper. Below, we respond to each of your comments and explain the revisions made. All changes are clearly highlighted in yellow in the revised manuscript, Please note that the page numbers referred to in this response letter correspond to the highlighted revision version of the revised manuscript.

1.      Point-by-point response to Comments and Suggestions for Authors

Comments 1: "The abstract contains the main points of the article but does not account for the research process, which includes several participants, as well as a summary of the findings with recommendations from the research findings."

Response 1: Thank you for pointing this out. We appreciate your feedback regarding the need to include a clearer account of the research process and to summarize the findings along with relevant recommendations in the abstract.

In response, we have revised the abstract to:

Provide a concise description of the research process (including the number of participants).

Summarize the key findings, including the identification of the high-performance traits and their distinction between early and late career stages.

Highlight the practical implications and recommendations that stem from our research findings, focusing on teacher training, selection, and development.

These changes can be found in Page 1, Abstract section, lines 10–22.

The updated abstract now reads as follows:

To promote enhanced well-being among children, collaboration among families, schools, and communities is paramount—a pressing concern in the global education sector. This necessitates that preschool teachers possess the necessary competencies for effective Family-Preschool Partnerships(FPP). This study explored the competencies necessary for Chinese kindergarten teachers to engage in FPP using behavioral event interviews with 30 participants. Thematic analysis identified key competency traits, and independent samples t-tests with Bonferroni correction compared collaboration competencies between outstanding and typical teachers, as well as across different career stages. Consequently, a comprehensive crisscrossing competency framework consisting of four quadrants was developed. This framework distinguishes between high-performance and general traits, as well as between stable and variable traits that may evolve across career stages. High-performance traits such as communication, expression, and relationship management should be prioritized in the training and recruitment of early childhood educators involved in FPP. In contrast, intrinsic qualities that foster successful FPP, such as child-orientation, should be cultivated early and sustained throughout a teacher's career. From a developmental perspective, this framework provides a crucial foundation for evaluating and training kindergarten teachers in the competencies essential for fostering effective FPP.

Comments 2: "Keywords – It is advisable to use acronyms for phrases that will appear throughout the article, such as: family-preschool partnerships = FPP."

Response 2: Thank you for this suggestion. We have incorporated the acronym "FPP" for "family-preschool partnerships" throughout the manuscript to improve consistency and readability.

This change has been applied consistently across the manuscript, starting from the abstract and introduction sections.

Comments 3: "According to the submitted article, there is no literature review, but on the other hand, the introduction to the work is very long. The introduction should be divided so that the article contains an introduction and a literature review. lines 26–153."

Response 3: Thank you for this valuable suggestion. I fully agree with your comment, and accordingly. We have divided the original introduction into two parts: the introduction and the theoretical framework and literature review.

The Introduction now focuses on the research background, the problem statement, and the significance and rationale of the study, with the aim of providing readers with a clear contextual foundation and the justification for conducting this research.

 This change can be found in lines 29–60 of the revised manuscript.

Updated text in the manuscript (Introduction):

While the involvement of families and communities in school education has often been overlooked or avoided in China, this issue is not unique to this context. Similar patterns have been observed in other countries, which can be attributed in part to the failure to recognize this topic as an integral component of every teacher's professional responsibilities (Epstein, 2018). Educational policy documents issued by the Chinese Ministry of Education (2023; 2024) highlight the importance of family-school-community cooperation, recognizing it as essential for fostering an environment conducive to childrens holistic development and well-being.

 Collaboration among families, educational institutions, and society in the upbringing of children represents a pioneering advancement in the educational ideology (Sun et al., 2023). Epstein's "Overlapping Spheres of Influence" theory (2011) provides a framework that emphasizes the importance of trust and mutual commitment in fostering a successful family-school partnership. Such collaboration not only supports children's academic achievements but also bolsters their overall well-being (Epstein, 2013; Mayer, 1994; Markström & Simonsson, 2017; Uludag, 2008).

Nevertheless, the current practice of collaboration between teachers and parents in children's education continues to face numerous challenges and obstacles. A significant factor contributing to this predicament is the deficiency in teachers' "professionalism" and "competency" regarding family-school partnerships (Denessen, et al., 2009; Willemse et al., 2018). In the actual implementation of collaborative efforts between teachers and parents, educators often lack sufficient competence to effectively guide and facilitate parental involvement in their children's learning (Epstein, 2013). Insufficient cognitive proficiency and a lack of relevant skills result in many teachers, especially those new to the profession, struggling to engage in fruitful communication and collaboration with their parents (de Bruïne et al., 2014; Willemse et al., 2015). Consequently, this deficiency in collaborative competence hampers teachers' ability to effectively co-educate children in concert with their parents and the broader community (Visković and Višnjić Jevtić 2017). Hence, it is imperative that we direct our research focus to the enhancement of teachers’ competency for family-school partnerships. This study aims to address this need by examining the competencies required for preschool teachers to promote the development of a high-quality early childhood education workforce and contribute to children’s overall development.

Meanwhile, the Theoretical background and literature review section is dedicated to discussing the existing theoretical frameworks, the current state of research in the field, identified research gaps, and the importance of addressing these gaps. It concludes by articulating the specific aims of the present study.

The section is structured into three parts:

  • The theoretical framework is discussed in lines 62–98,
  • The review of current research and identification of research gaps are presented in lines 99–135, and
  • The research objectives are clearly stated in Lines 136–148.

Comments 4: lines 75-80 : Correct wording”

Response 4: Thank you very much for pointing this out. We sincerely apologize for the confusion caused by our original wording due to language expression issues. We have revised and polished this part to improve clarity and ensure accurate expression. We hope the updated version better conveys our intended meaning.

Changes can be found in lines 110–117of the revised manuscript.

Updated text in the manuscript (lines 110–117):

However, research on FPPC remains limited, with existing studies primarily exploring these competencies from the perspectives of specific stakeholder roles, as summarized in Table 1. This table synthesizes key FPPC frameworks, illustrating the varied conceptualizations based on different professional and contextual standpoints. For example, Westergård (2013) emphasizes relational, communicative, and contextual competencies within the parent–teacher interaction context in Norway. Similarly, Vuorinen et al. (2014) highlight the importance of social and communication skills among preschool teachers in Sweden.

Comments 5: "Figures 1 and 2 are not sharp and do not fit into the final version of the article."

Response 5: Thank you for bringing this to our attention. We have replaced Figures 1 and 2 with higher resolution images and ensured that they fit appropriately within the layout of the manuscript. Specifically:

  • Figure 1 is located on Page 3 of the revised manuscript.
  • The original Figure 2 has been renumbered as Figure 3, and now appears on Page 11.

Please review the Figures in the revised manuscript.

Comments 6: “Starting from line 117, there is an explanation of partnerships between the kindergarten and parents. There is no mention, for example, that nowadays communication with parents can be conducted via digital platforms such as WhatsApp for the benefit of the child's advancement and well-being.”

Response 6: Thank you for this insightful suggestion. We have added a mention of the role of digital platforms, such as WeChat and Ding Talk, in facilitating communication between kindergarten teachers and parents, and we agree that it should be included in the manuscript.

These revisions can be found in the revised manuscript on Page 2, lines 91–94.

Updated text in the manuscript:

In recent years, the development of digital communication has further strengthened this collaboration. For example, digital platforms such as WeChat and Ding Talk have become valuable tools for enhancing communication between teachers and parents, enabling more timely, accessible, and continuous interaction.

Comments 7: “Methods and Materials - A diagram describing the research process should be attached to this chapter, which will help the reader to concisely understand the research process in a consistent manner.”

Response 7: Thank you sincerely for this helpful suggestion. We fully agree that including a diagram in the Methods and Materials section can facilitate readers’ understanding of the research procedure. Accordingly, we have added a research process diagram to this chapter (Figure 2, Page 5) to present the methodological steps in a clear and consistent manner. We hope this visual addition enhances the transparency and accessibility of the study’s design. The research process is illustrated in the diagram below.

Figure 2. Research process diagram.

Comments 8: “Results - is written in detail, but unfortunately due to technical problems it is very difficult to read. See for example lines 397-402, 412-414, 437-439, 462-464…….”

Response 8 Thank you very much for pointing this out. We truly appreciate your careful reading and constructive feedback. We acknowledge that the technical formatting and sentence structure in the original Results section may have affected the clarity and readability. In response, we have carefully revised this section to improve its coherence, simplify complex expressions, and ensure that the findings are communicated clearly and logically. The revisions can be found throughout the revised manuscript, particularly on lines 390–400,410–414, 435–442, and 464-469. We hope the updated version enhances the readability and allows the results to be understood more easily. These revisions can be found on Pages 12–14 of the revised manuscript.

We have carefully revised lines 390–400, which now read as follows:

Teacher Yu, a teacher with over 15 years of experience, shared the following in the interview:

Through years of experience, I have come to understand the importance of effective relationship management. In our classroom, communication between the three teachers is vital. For instance, if an incident occurs, such as a child falling, my co-teacher and caregiver will immediately inform me. I always prioritize discussing such incidents with the child's mother first. If I’m unaware of the situation when parents inquire, they may feel that the teachers are neglecting their child. That’s why communication and cooperation among teachers are essential. We also remind each other of various tasks to ensure nothing is forgotten. By maintaining a consistent approach when communicating with parents, we can take the initiative in these discussions(Teacher Yu, Personal communication, February 2025).” This change can be found on Page 12, lines 390–400 of the revised manuscript.

We have carefully revised lines 412–414, which now read as follows:

Later in her career, Teacher Guo emphasized that family–preschool partnerships should be viewed as a long-term, interactive process. This involves the continuous exchange of information, emotions, expectations, and ideas—an approach consistent with Adams and Christenson’s (2000) perspective. As teacher Guo explained during the interview:” Updated text in the manuscript(Page 13, lines 410–414).

We have also revised the original lines 437–439 to improve clarity and readability. The revised text now reads as follows:

“Many teachers face challenges rooted in their own family backgrounds and limited professional experience. These difficulties are often compounded by the lack of systematic training in family education during their teacher preparation. As a result, they may have limited exposure to diverse family structures and parenting philosophies, which hinders their competence to engage effectively with families. When faced with unique family-related issues, this knowledge gap can leave them feeling powerless”.Updated text in the manuscript(Page 13, lines 435–442).

We have also revised the original lines 462–464 for greater clarity and fluency. The updated version now reads as follows:

As Teacher Dong noted during the interview:

When you genuinely care for a parent's child, with true devotion and a commitment to guiding and accompanying the child at every stage in kindergarten,you foster a sense of love and respect that both the child and the parent can feel, leading to a positive perception of the teacher (Teacher Dong, personal communication, January 2025). “Updated text in the manuscript(Page 14, Lines 464–469).

Comments 9: Discussion - It is necessary to shorten and, above all, integrate and conduct a more extensive discussion between the review and the findings.

 Response 9: Thank you very much for this insightful suggestion. We fully agree that the original version of the discussion was overly lengthy and lacked sufficient integration between the literature review and our empirical findings.

In the revised manuscript, we have undertaken the following changes to address your comment:

  • Shortened and streamlined the discussion: We removed redundant explanations and consolidated overlapping content to improve conciseness and readability. We removed redundant phrases and repetitive descriptions, reducing the word count from 804 to approximately 500 words. (see revised Discussion, Page 15, Lined 514–561).
  • Stronger structure and clarity: The revised discussion now follows a three-part structure:
    (1) Paragraph 1: Summary of research aims and key findings;
    (2) Paragraph 2-3: Dialogue with previous studies, organized into two paragraphs based on the categorization of FPPC traits (i.e., performance relevance and trait stability);
    (3) Paragraph 4: Practical applications of the crisscrossing framework.

To improved logical flow, we employed clear transition markers such as First,  Additionally, and To translate these findings into practice to ensure smooth progression and coherence.

  • Highlighting Key Findings: In the revised first paragraph of the discussion, we emphasized the primary contributions of this study to the conceptual understanding of Family–Preschool Partnership Competence (FPPC). Specifically, the revision underscores how our work refines the content structure of FPPC and breaks through the limitations of prior static frameworks.

Updated text in the manuscript(lines 514–518):

Previous researches have indicated that competencies serve as a vital link between educational preparation and job requirements (van der Klink & Boon, 2002). This study employs a mixed-methods, integrating qualitative and quantitative research techniques, to develop a comprehensive crisscrossing framework of FPPC. The framework not only distinguishes key high-performance traits from general traits pertinent to FPP, but also conducts a comparative analysis of these traits based on early career stage and late career stage, categorizing them as stable traits or variable traits. It is designed to aid in the training, development, and selection of teachers.

  • Integrated findings with prior literature: We revised key paragraphs to explicitly connect our results with the theoretical and empirical work cited in the literature review. For instance, when discussing communication and emotional regulation traits, we now directly relate them to findings by Westergård (2013), Musset (2010), and Klassen et al. (2020), thereby deepening the dialogue between empirical results and existing frameworks.

Updated text in the manuscript(lines 519–544):

First, the study identified several high-performance traits, such as child-orientation, emotional regulation, communication, and expression skills, directly influencing the effectiveness of family preschool partnerships. These results align with previous research indicating that relational and communication skills are essential for collaboration between teachers and parents (Westergård 2013; Musset 2010). Given that these traits are cultivable, yet not universally present, schools should emphasize the precise identification of high-performance traits within their teacher management systems and deliver targeted training accordingly. Furthermore, the trainability of these traits underscores their potential for early identification and developmental interventions through structured assessments and interactive learning tools, including situational judgment tasks, mentoring, and video-based reflection (Rockstuhl et al., 2015; De Coninck et al., 2018).

Additionally, the analyses revealed that certain traits, such as reflective learning, positive attribution, and emotional stability, tend to remain relatively stable across different career stages. Since implicit trait policies are rooted in core beliefs and personality characteristics shaped by early socialization processes (Klassen et al., 2020), their transformation often requires substantial time and effort. However, current teacher support and training systems often focus on superficial skill development, neglecting the deeper transformation of intrinsic traits (Wu 2016). This may also reflect developmental stagnation in certain regions of China due to inadequate support for kindergarten teachers, rather than an inherent attribute of competence. Furthermore, foundational knowledge related to parenting and child development, both physical and psychological, was consistent between early- and late-career teachers. The findings contrast with Xia's (2023) observation that teachers' competence in fostering FPP varies significantly across different developmental stages. A potential explanation lies in the enduring impact of pre-service theoretical training on Chinese kindergarten teachers. coupled with insufficient post-service professional development to update their knowledge base (Wu, 2016).

  • Added synthesis and practical implications: The final paragraph outlines implications of our crisscrossing model for selection, training, and development in early childhood education, ensuring applied relevance. These revisions help ensure that the discussion is analytically richer, more connected to the reviewed literature, and practically meaningful for readers.

Updated text in the manuscript(lines 545–561):

To translate these findings into practice, our four-quadrant model can be applied to teacher training, evaluation, and professional development by differentiating competency traits based on performance orientation and variability. Quadrant I traits (high-performance and variable), such as empathy and communication skills, should be prioritized during recruitment and training through interactive skill instruction and feedback mechanisms. Quadrant II traits (high-performance, stable), such as emotional regulation and reflective thinking, are best identified through situational interviews during the teacher selection process (Lievens & Motowidlo, 2016) and require ongoing support to maintain stability. The third quadrant encompasses intrinsic tendencies, such as initiative and achievement motivation. Although these traits are not always overtly expressed in daily interactions, they can foster professional growth and resilience. Finally, Quadrant IV comprises foundational knowledge and basic skills; these traits (e.g., observation and understanding, differentiated teaching ability) tend to be developed through systematic training and practical work experience rather than relying solely on an individual's predisposition or short-term experience. Therefore, it is important to pay attention to teachers' performance in FPP activities in their daily work and to continuously monitor their growth and development in these areas.

Comments 10:Conclusion - Well written and in a findings format. Conclusions and limitations of the study should be separated into 2 separate subsections.”

 Response 10: Thank you for your positive evaluation of the writing quality of the Conclusion section. We appreciate your suggestion to improve the structure by separating the “Conclusion” and “Limitations and Future Research” into two distinct subsections. We fully agree that this enhances clarity and aligns with academic conventions. Therefore, we have revised the manuscript accordingly.

 The revised section now appears as two clearly labeled parts: (1) Conclusion, which summarizes the key findings and theoretical contributions of the study, and (2) Limitations and Future Research, which outlines methodological constraints and proposes directions for future inquiry.

These changes can be found on pages 15–16, lines 562-602 of the revised manuscript, with all modifications highlighted in yellow.

Comments 11: ”Limitations and Future Research - The text should be reduced by half, and the fact that this study included only 30 kindergarten teachers, which is a very small percentage of all kindergarten teachers in China and in relation to the size of the population, should be emphasized.” and “Also missing is a recommendation for further research on the subject, such as similar research with private kindergarten teachers and/or those not employed in public and/or government settings.”

 Response 11: Thank you very much for your constructive and detailed comments. We agree that the original Limitations and Future Research section was overly lengthy and needed to more clearly acknowledge the limited generalizability of the sample. In response, we have revised this section by reducing its length by approximately 50% (reduce 394 words to 202 words)and explicitly highlighting that the sample size—30 kindergarten teachers from central China—represents a very small proportion of the national preschool teacher population.

Additionally, as per your suggestion, we have added a more targeted recommendation for future research to examine FPPC among teachers working in private or non-governmental kindergarten settings. We also included suggestions for expanding to other regions and adopting longitudinal research designs to examine developmental trajectories. These changes are located on page 16, lines 583–601 of the revised manuscript and have been marked in yellow for clarity.

Updated text in the manuscript(Limitations and Future Research):

Although this study offers a comprehensive understanding of kindergarten teachers' FPPC using the trait-performance quadrant framework, several limitations warrant acknowledgment. The qualitative research design permits in-depth insights into the developmental logic and trait differences of the FPPC. However, the sample comprised only 30 kindergarten teachers from central China, representing a small and non-representative segment of early childhood educators nationwide. This limitation restricts the generalizability of the findings to other regions. Additionally, despite coder training and cross-verification, interpretation subjectivity may not have been completely eliminated. The cross-sectional nature of this study further constrains our ability to assess how FPPC traits evolve over time or in response to varying career stages and situational demands.

Future research should prioritize the use of large-scale, diverse samples to validate the proposed capability framework, encompassing teachers from more private and non-public institutions. Longitudinal designs would be particularly beneficial for tracking trait expression changes across different career stages. Furthermore, the development of standardized, context-sensitive assessment tools could facilitate the quantification of Family and FPPC traits and inform recruitment and training programmes. Finally, subsequent studies could investigate whether the identified capabilities can be transferred to other collaborative environments, such as school-community or school-healthcare partnerships, to further advance the field of early education.

Comments 12: “Line 790 unnecessary space delete.” And “Lines 794, 797 & 812 right-aligned and reduced spacing.”

Response 12: Thank you for pointing out these formatting inconsistencies. We have carefully reviewed and corrected the formatting issues you identified:

  • The unnecessary space at Line 790 has been deleted.
  • Lines 794, 797, and 812 have been right-aligned consistently with the rest of the manuscript, and the excessive line spacing has been reduced.

These changes help improve the visual consistency and readability of the manuscript. All corrections have been clearly marked in yellow in the revised file.

Updated text in the manuscript (Pages 19-20, lines 740-758):

Yu, Q. F., Chen, J. Y., & Song, H. (2022). Construction of teacher-parent cooperation competence indicators based on the Delphi method. Journal of Teacher Education Research, 34(6), 44-52. https://kns.cnki.net/kcms/detail/detail.aspx?dbname=CJFD2022&filename=GDSZ202206007&dbcode=CJFD

Yuan, K. M., Zhou, X. R., & Ye, P. Q. (2021). Research on the competence model of Home-school Cooperation among primary and secondary school teachers. China Educational Technology, 06, 98-104. https://kns.cnki.net/kcms/detail/detail.aspx?dbname=CJFD2021&filename=ZDJY202106013&dbcode=CJFD

Xu, J. P., & Zhang, H. C. (2006). A competency model of primary and secondary school teachers: A behavioral event interview study. Educational Research, 01, 57-61+87. https://kns.cnki.net/kcms/detail/detail.aspx?dbname=CJFD2021&filename=JYXX202110005&dbcode=CJFD

Final note: We sincerely thank Reviewer 2 for the time and effort dedicated to reviewing our manuscript. Your constructive comments have significantly contributed to improving the overall clarity, rigor, and scholarly quality of our work. We particularly appreciate your attention to formatting precision and the emphasis on integrating discussion and findings, which reminded us of the high standards required for SSCI journal submissions.

In response, we have carefully revised the manuscript according to all your suggestions. Specifically, we shortened and restructured the discussion section to enhance logical flow and coherence; we separated the conclusion and limitations into distinct subsections; and we corrected all formatting issues, including spacing and alignment inconsistencies. Furthermore, we have thoroughly polished the manuscript for grammar and style to ensure the writing meets the standards of academic English. Detailed revisions can be found in the tracked changes version of the manuscript.

We are grateful for your valuable insights and hope that the revised manuscript now meets your expectations.

Reviewer 3 Report

Comments and Suggestions for Authors

The introduction effectively sets the stage for a significant study, highlighting the importance of family-school partnerships in China and the global relevance of teacher competencies. However, the introduction needs improvement. It's a bit hard to follow and some parts are repetitive.

  • Please merge the repeated points about the importance of family-school partnerships and the challenges of teacher competency to create a more concise introduction.
  • Adding a clear definition of key terms like "crisscrossing" would help. For example, add a sentence after line 61 defining "crisscrossing": for example, "This study conceptualizes FPPC as a 'crisscrossing' framework, where teachers' knowledge, skills, and attitudes intersect with parental and institutional efforts to support child development."
  • The objectives can be made stronger by briefly explaining why comparing different types of teachers and career stages is important.
  • Please provide a brief explanation of the sampling process (lines 188-195). For instance: "This study employed purposive sampling to select preschool teachers with relevant experience in family-preschool partnerships, followed by snowball sampling to expand the sample through participant referrals. A total of 30 preschool teachers from ...."
  •  The demographic details are helpful but could be presented more systematically for instance in a table or with additional context (e.g., gender, teaching background).
  • In the Findings section, the paragraph is dense, with long lists of competency traits that make it hard to follow. Readers may struggle to track which traits are significant vs. non-significant or their relative importance. Please break the paragraph into smaller sections or use bullet points/subheadings to organise the findings.
  • The discussion section could be improved by better organizing it, making it clearer, and adding more depth. Include clear transitions to link findings, comparisons with other studies, and practical applications.

Thank you! 

Comments on the Quality of English Language

The English could be improved to more clearly express the research.

Author Response

Thank you for your thoughtful and constructive comments on our manuscript. We greatly appreciate the time and effort you put into reviewing our work. Your feedback has been incredibly valuable, and we agree with many of your points. After careful consideration and discussion within our team, we have made substantial revisions to the manuscript in response to your suggestions. Below, we provide a point-by-point response to each of your comments. Please note that the page numbers referred to in this response letter correspond to the highlighted revision version of the revised manuscript.

1.      Point-by-point response to Comments and Suggestions for Authors

Comments 1: "The introduction effectively sets the stage for a significant study, highlighting the importance of family-school partnerships in China and the global relevance of teacher competencies. However, the introduction needs improvement. It's a bit hard to follow and some parts are repetitive." and “Please merge the repeated points about the importance of family-school partnerships and the challenges of teacher competency to create a more concise introduction.”

Response 1: We appreciate your feedback and agree that the introduction could be clearer and more concise. We fully agree with your observation that the original introduction contained overlapping statements regarding the importance of family-school partnerships and the challenges related to teacher competencies.

To address this issue, we have divided the original introduction into two parts: the introduction and the theoretical framework and literature review. And we carefully revised the introduction by merging repetitive content, streamlining the narrative, and improving the logical flow. Specifically, we combined the discussion of policy developments, theoretical underpinnings (e.g., Epstein's framework), and the teacher competency gap into a more integrated and concise exposition. The revised version more clearly distinguishes between background context, research gaps, and study objectives. The Introduction now focuses on the research background, the problem statement, and the significance and rationale of the study, to provide readers with a clear contextual foundation and the justification for conducting this research.

Revised Introduction (Pages 2-3, lines 30-60):

While the involvement of families and communities in school education has often been overlooked or avoided in China, this issue is not unique to this context. Similar patterns have been observed in other countries, which can be attributed in part to the failure to recognize this topic as an integral component of every teacher's professional responsibilities (Epstein, 2018). Educational policy documents issued by the Chinese Ministry of Education (2023; 2024) highlight the importance of family-school-community cooperation, recognizing it as essential for fostering an environment conducive to children’s holistic development and well-being.

 Collaboration among families, educational institutions, and society in the upbringing of children represents a pioneering advancement in the educational ideology (Sun et al., 2023). Epstein's "Overlapping Spheres of Influence" theory (2011) provides a framework that emphasizes the importance of trust and mutual commitment in fostering a successful family-school partnership. Such collaboration not only supports children's academic achievements but also bolsters their overall well-being (Epstein, 2013; Mayer, 1994; Markström & Simonsson, 2017; Uludag, 2008).

Nevertheless, the current practice of collaboration between teachers and parents in children's education continues to face numerous challenges and obstacles. A significant factor contributing to this predicament is the deficiency in teachers' "professionalism" and "competency" regarding family-school partnerships (Denessen, et al., 2009; Willemse et al., 2018). In the actual implementation of collaborative efforts between teachers and parents, educators often lack sufficient competence to effectively guide and facilitate parental involvement in their children's learning (Epstein, 2013). Insufficient cognitive proficiency and a lack of relevant skills result in many teachers, especially those new to the profession, struggling to engage in fruitful communication and collaboration with their parents (de Bruïne et al., 2014; Willemse et al., 2015). Consequently, this deficiency in collaborative competence hampers teachers' ability to effectively co-educate children in concert with their parents and the broader community (Visković and Višnjić Jevtić 2017). Hence, it is imperative that we direct our research focus to the enhancement of teachers’ competency for family-school partnerships. This study aims to address this need by examining the competencies required for preschool teachers to promote the development of a high-quality early childhood education workforce and contribute to children’s overall development.

This revision eliminates redundancy and makes the introduction more concise and easier to follow.

Comments 2: " Adding a clear definition of key terms like "crisscrossing" would help. For example, add a sentence after line 61 defining "crisscrossing": for example, "This study conceptualizes FPPC as a 'crisscrossing' framework, where teachers' knowledge, skills, and attitudes intersect with parental and institutional efforts to support child development."

Response 2: Thank you for your thoughtful suggestion regarding the need to define the term "crisscrossing." We completely agree that providing a clear and precise definition would enhance readers’ understanding of the core framework used in this study.

After careful consideration, we have added a definition of “crisscrossing” the first time the term appears in the description of research objectives. The revised sentence now reads:

Objective 3: To develop a sustainable  crisscrossing FPPC framework. ( A dual-axis design that categorizes competencies along two dimensions: the horizontal axis distinguishes between traits of high-performing and general, while the vertical axis classifies these traits by their degree of variability.) This framework aims to better support preschool teachers’ development and retention.

This revision appears on page 4, lines 144–148 in the revised manuscript.

We believe this clarification not only supports better reader comprehension but also highlights the conceptual innovation of our study. Thank you again for prompting this important addition.

Comments 3: "The objectives can be made stronger by briefly explaining why comparing different types of teachers and career stages is important."

Response 3: We appreciate your comment regarding the clarification of the research objectives. We agree that providing a brief explanation of why comparing different types of teachers and career stages is important would enhance the rationale behind the objectives. As suggested, we have updated the objectives section to explain the significance of comparing early-career and late-career teachers: An addition for Objective 2 clarifies that these comparisons help us explore how these competencies evolve and develop with experience.

Revised text (Page 4, Lines 141-143):

(2) Objective 2: To identify notable differences in competencies between early and late career stages in family preschool partnership work, exploring how these competencies evolve and develop with experience.

This revision strengthens the objectives by explaining the importance of comparing teacher career stages, which adds depth to the research focus.

Comments 4: "Please provide a brief explanation of the sampling process (lines 188-195). For instance: "This study employed purposive sampling to select preschool teachers with relevant experience in family-preschool partnerships, followed by snowball sampling to expand the sample through participant referrals. A total of 30 preschool teachers from ...."

Response 4: Thank you for this suggestion. We have now included a more detailed explanation of the sampling process, as recommended. We clearly describe the use of purposive sampling and snowball sampling, which helped us select a targeted sample of preschool teachers with relevant experience.

Here’s the updated explanation(Page 6, 186-189):

This study employed purposive sampling to select preschool teachers with relevant experience in FPP, followed by snowball sampling to expand the sample through participant referrals. A total of 30 preschool teachers from four provinces and municipalities in the east-central region of China were recruited for this study.

Comments 5: “The demographic details are helpful but could be presented more systematically for instance in a table or with additional context (e.g., gender, teaching background).”

Response 5: Thank you very much for this insightful recommendation. We agree that presenting the demographic details in a more structured and accessible format would enhance the clarity and readability of the manuscript. In response, we have created a new table (Table 2) to systematically present key participant information, including gender, teaching years of experience, kindergarten type (public vs. private), and teaching location.

Please review Table 2 in the revised manuscript on page 6.

Comments 6: “In the Findings section, the paragraph is dense, with long lists of competency traits that make it hard to follow. Readers may struggle to track which traits are significant vs. non-significant or their relative importance. Please break the paragraph into smaller sections or use bullet points/subheadings to organise the findings.”

Response 6: Thank you very much for this valuable suggestion. We fully agree that the original Findings section was too dense, with extensive text that may have made it difficult for readers to clearly distinguish the types and significance of competency traits.

In response, we have substantially revised this section to improve clarity and readability. Specifically:

  • We reorganized the content based on the four-quadrant FPPC framework, which distinguishes traits by performance relevance (high-performing vs. general) and stability (stable vs. variable).
  • Within each quadrant, we added subheadings to guide readers through the structure.
  • We used bullet points under each quadrant to clearly list the corresponding competency traits.
  • We revised and condensed the descriptions to highlight the nature, developmental characteristics, and practical implications of each trait, while also reducing redundancy and overly complex sentence structures.

These changes now allow readers to better track which traits are critical, which are malleable or stable, and how they align with the teachers' professional development.

The revised content appears in the Findings section, pages 11-14, lines 348–512 of the revised manuscript.

Comments 7: “The discussion section could be improved by better organizing it, making it clearer, and adding more depth. Include clear transitions to link findings, comparisons with other studies, and practical applications.”

Response 7: Thank you very much for your thoughtful and constructive feedback. We fully agree with your suggestions regarding the clarity, structure, and depth of the Discussion section. Your comments prompted us to carefully reflect on how to better integrate the findings with the existing literature and emphasize their practical implications.

In response, we made the following substantial revisions:

l   Stronger structure and clarity: The revised discussion now follows a three-part structure:
(1) Paragraph 1: Summary of research aims and key findings;
(2) Paragraph 2-3: Dialogue with previous studies, organized into two paragraphs based on the categorization of FPPC traits (i.e., performance relevance and trait stability);
(3) Paragraph 4: Practical applications of the crisscrossing framework.

To improve logical flow, we employed clear transition markers such as First, Additionally, and To translate these findings into practice to ensure smooth progression and coherence.

l   Highlighting Key Findings: In the revised first paragraph of the discussion, we emphasized the primary contributions of this study to the conceptual understanding of Family–Preschool Partnership Competence (FPPC). Specifically, the revision underscores how our work refines the content structure of FPPC and breaks through the limitations of prior static frameworks.

Updated text in the manuscript(lines 514–518):

Previous researches have indicated that competencies serve as a vital link between educational preparation and job requirements (van der Klink & Boon, 2002). This study employs a mixed-methods, integrating qualitative and quantitative research techniques, to develop a comprehensive crisscrossing framework of FPPC. The framework not only distinguishes key high-performance traits from general traits pertinent to FPP, but also conducts a comparative analysis of these traits based on early career stage and late career stage, categorizing them as stable traits or variable traits. It is designed to aid in the training, development, and selection of teachers.

l   Integrated findings with prior literature: We revised key paragraphs to explicitly connect our results with the theoretical and empirical work cited in the literature review. For instance, when discussing communication and emotional regulation traits, we now directly relate them to findings by Westergård (2013), Musset (2010), and Klassen et al. (2020), thereby deepening the dialogue between empirical results and existing frameworks.

Updated text in the manuscript(lines 519–544):

First, the study identified several high-performance traits, such as child-orientation, emotional regulation, communication, and expression skills, directly influencing the effectiveness of family preschool partnerships. These results align with previous research indicating that relational and communication skills are essential for collaboration between teachers and parents (Westergård 2013; Musset 2010). Given that these traits are cultivable, yet not universally present, schools should emphasize the precise identification of high-performance traits within their teacher management systems and deliver targeted training accordingly. Furthermore, the trainability of these traits underscores their potential for early identification and developmental interventions through structured assessments and interactive learning tools, including situational judgment tasks, mentoring, and video-based reflection (Rockstuhl et al., 2015; De Coninck et al., 2018).

Additionally, the analyses revealed that certain traits, such as reflective learning, positive attribution, and emotional stability, tend to remain relatively stable across different career stages. Since implicit trait policies are rooted in core beliefs and personality characteristics shaped by early socialization processes (Klassen et al., 2020), their transformation often requires substantial time and effort. However, current teacher support and training systems often focus on superficial skill development, neglecting the deeper transformation of intrinsic traits (Wu 2016). This may also reflect developmental stagnation in certain regions of China due to inadequate support for kindergarten teachers, rather than an inherent attribute of competence. Furthermore, foundational knowledge related to parenting and child development, both physical and psychological, was consistent between early- and late-career teachers. The findings contrast with Xia's (2023) observation that teachers' competence in fostering FPP varies significantly across different developmental stages. A potential explanation lies in the enduring impact of pre-service theoretical training on Chinese kindergarten teachers. coupled with insufficient post-service professional development to update their knowledge base (Wu, 2016).

l   Added synthesis and practical implications: The final paragraph outlines implications of our crisscrossing model for selection, training, and development in early childhood education, ensuring applied relevance. These revisions help ensure that the discussion is analytically richer, more connected to the reviewed literature, and practically meaningful for readers.

Updated text in the manuscript(lines 545–561):

To translate these findings into practice, our four-quadrant model can be applied to teacher training, evaluation, and professional development by differentiating competency traits based on performance orientation and variability. Quadrant I traits (high-performance and variable), such as empathy and communication skills, should be prioritized during recruitment and training through interactive skill instruction and feedback mechanisms. Quadrant II traits (high-performance, stable), such as emotional regulation and reflective thinking, are best identified through situational interviews during the teacher selection process (Lievens & Motowidlo, 2016) and require ongoing support to maintain stability. The third quadrant encompasses intrinsic tendencies, such as initiative and achievement motivation. Although these traits are not always overtly expressed in daily interactions, they can foster professional growth and resilience. Finally, Quadrant IV comprises foundational knowledge and basic skills; these traits (e.g., observation and understanding, differentiated teaching ability) tend to be developed through systematic training and practical work experience rather than relying solely on an individual's predisposition or short-term experience. Therefore, it is important to pay attention to teachers' performance in FPP activities in their daily work and to continuously monitor their growth and development in these areas.

Final Note: We sincerely thank Reviewer 3 for the time, attention, and insightful comments provided throughout the review process. Your thoughtful feedback has been instrumental in enhancing the overall quality and clarity of our manuscript. We especially appreciate your suggestions to simplify and streamline the introduction, provide a clearer explanation of the sampling procedure, restructure the Findings section with improved organization, and deepen the connection between our findings and the literature in the Discussion. These points reminded us of the importance of precision, coherence, and critical engagement in SSCI-level academic writing.

In response, we have thoroughly revised the manuscript. Specifically, we:

  • Simplified and reorganized the Introduction to remove redundancies and clarify the definition and rationale for the crisscrossing framework;
  • Added a more systematic description of the sample and sampling procedures, along with a demographic table;
  • Reformatted the Findings section using quadrant-based headings and bullet points to improve readability and highlight key competencies and their traits;
  • Restructured the Discussion into three coherent parts—summary of findings, dialogue with prior research, and practical implications—with clearer transitions and reduced wordiness.
  • Revised the Conclusion;
  • Finally, the entire manuscript was professionally polished to improve grammatical accuracy, language fluency, and formatting consistency. Detailed revisions can be found in the tracked changes version of the manuscript.

We are sincerely grateful for your detailed and constructive review and hope the revised manuscript now meets the standards expected for publication.

Round 2

Reviewer 2 Report

Comments and Suggestions for Authors

The authors of the article revised the article according to the comments and made the necessary changes
The article is now more accurate and worthy of publication
Regards,
The Reviewer